# Transit Timing Variations of Exoplanet WASP-4b: Decrease in the Orbital Period

**AI R&D, Gaston Longhitano** [1]**, Avi Shporer** [2]**, Iddo Drori** [3,4,5]

[1] Boston University, [2] MIT, [3] Yeshiva University, [4] Tel Aviv University, [5] Stanford University

## Abstract

Close-in giant planets provide rare laboratories for measuring tidal dissipation in stars through long-baseline transit timing. We analyze four TESS sectors of WASP-4b photometry (Sectors 2, 28, 29, and 69), measure per-transit mid-times with a limb-darkened Mandel–Agol model, and combine these with twelve legacy, non-TESS timings to extend the baseline back to 2008. A quadratic ephemeris is decisively favored over a constant-period model ($\Delta$BIC $\approx$ 313), yielding a negative period derivative of $\dot{P}$ = -13.77 $\pm$ 0.77 ms yr$^{-1}$ and a characteristic orbital decay timescale of $P/|\dot{P}| \approx 8.4 \times 10^6$ yr. Robustness checks (sector jackknifes, timing-error inflation, and SAP vs. PDCSAP photometry) leave the preference for a quadratic ephemeris intact. The simplest interpretation is tidal orbital decay, though slow line-of-sight acceleration (Rømer effect) or additional companions cannot be fully excluded without complementary radial-velocity monitoring.

## 1 Introduction

Hot Jupiters — large gas-giant planets on short orbital periods of only a few days — that skim their host stars offer a natural laboratory to test theories of tidal dissipation. Long, precise baselines of mid-times $T_{\mathrm{mid}}$ of transits (when the planet moves in front of its star and blocks a small fraction of the star light) allow us to search for secular departures from a constant orbital period. A negative period derivative ($\dot{P} < 0$) is an expected consequence of orbital decay if the stellar tidal quality factor $Q'_\star$ is sufficiently small, whereas other mechanisms—apsidal precession, light-time (Rømer) acceleration, or unseen companions—can also imprint curvature in the observed-minus-calculated (O–C) diagram comparing the observed mid-transit times to the expected times assuming a constant period (also referred to as a linear transit ephemeris).

WASP-4b is a well-studied hot Jupiter ($P \simeq 1.34$ d) that has displayed early transits relative to constant-period predictions since the start of the NASA Transiting Exoplanet Survey Satellite (*TESS*) space mission [1]. These anomalies were emphasized by Bouma et al. [2] and followed up by multiple authors who assembled large timing catalogs [3, 4, 5, 6]. The most recent work [7] interprets the curvature as tidal orbital decay.

This work provides a reproducible re-analysis focused on four *TESS* sectors (2, 28, 29, 69) combined with non-*TESS* timings from the literature. Our contributions are a transparent timing pipeline with uncertainty propagation from transit morphology, as shown in Figure 1, and an O–C diagram including both literature and *TESS* timings after subtracting a linear ephemeris, as shown in Figure 2.

## 2 Methods

**Photometry and quality control.** We analyze publicly available *TESS* SPOC PDCSAP light curves [8] for Sectors 2, 28, 29, and 69. We retain only cadences with quality flag set to

1st Open Conference on AI Agents for Science (agents4science 2025).

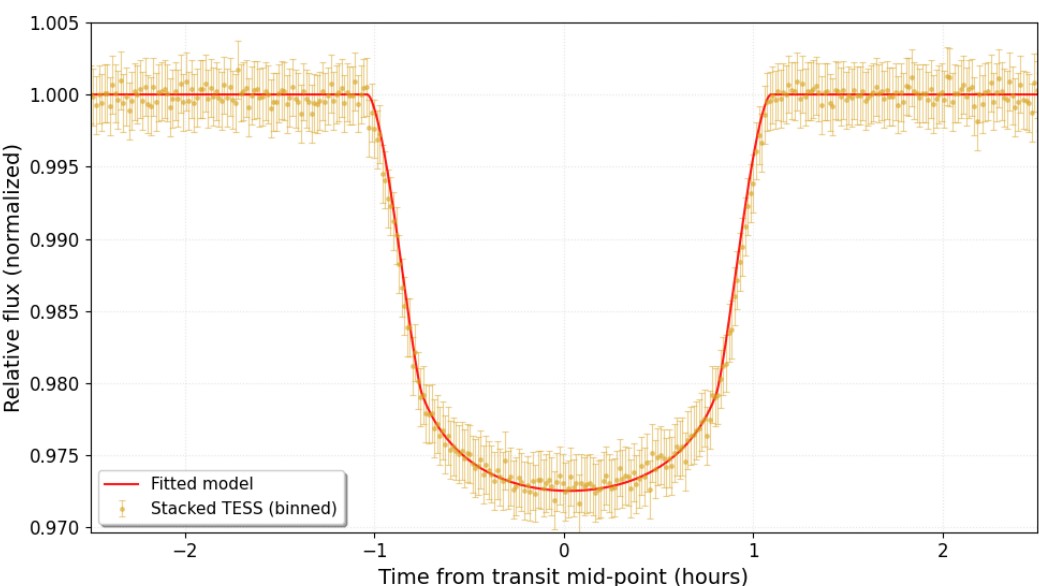

Figure 1: **Stacked *TESS* transit with fitted model (solid).** Abscissa in hours; ordinate is normalized relative flux.

zero (`QUALITY=0`) thereby rejecting measurements of poor quality. Time stamps are converted to $\mathrm{BJD_{TDB}}$ using `TIME` $+\,2457000$. Each predicted transit event is windowed by $\pm 0.12$ d. Within a window, we fit and divide out a linear out-of-transit (OOT) baseline using points with $|t - T_{\mathrm{pred}}| > 0.07$ d, then normalize by the OOT median, thereby converting the data to relative flux, relative to the OOT brightness (see Figure 1).

**Transit model and per-transit timing.** Transit shapes are modeled with a quadratic limb-darkened law Mandel–Agol profile [9]. We first build a high-S/N stacked transit aligned on an initial ephemeris to estimate the global morphology parameters $\theta = (p,\, a/R_\star,\, b,\, u_1,\, u_2)$, including the planet to star radius ratio, orbital semi-major axis normalized by the star's radius, the transit impact parameter (distance of the transit chord from the center of the star in units of the star's radius), and the two quadratic limb-darkening law coefficients. Holding $\theta$ fixed for individual transit events, we fit $(T_{\mathrm{mid}},\, a_0,\, a_1)$, where $a_0 + a_1(t - T_{\mathrm{pred}})$ captures the local baseline. The stacked transit and best-fitting model are shown in Figure 2.

**Uncertainty propagation from morphology.** To avoid underestimating timing errors, we propagate uncertainty in $\theta$ into $\sigma_T$ using a finite-difference Jacobian $J = \partial T_{\mathrm{mid}}/\partial \theta$ and the covariance $C_\theta$ from the stacked fit, inflating the per-transit timing variance as

$$\sigma_{T,\mathrm{tot}}^2 \simeq \sigma_{T,\mathrm{meas}}^2 + J\, C_\theta\, J^\top. \tag{1}$$

This closes the common gap between "fixed-shape" timing and realistic errors.

**Literature timings.** We include twelve non-*TESS* timings from Southworth et al. [3], Wilson et al. [10], Gillon et al. [11], Sanchis-Ojeda et al. [12], Huitson et al. [13], converted and/or verified to $\mathrm{BJD_{TDB}}$. These extend the 5 year *TESS* data baseline (2018–2023) by a decade, back to 2008, as shown in Table 1.

**Ephemerides and model selection.** Let $E$ be the integer epoch. We fit a linear ephemeris,

$$T(E) = T_0 + P\,E, \tag{2}$$

and a quadratic ephemeris,

$$T(E) = T_0 + P\,E + \frac{1}{2}\,Q\,E^2, \tag{3}$$

by weighted least squares to the combined timings. We compare models using the Bayesian Information Criterion,

$$\text{BIC} = \chi^2 + k \ln N, \tag{4}$$

which penalizes extra parameters. For interpretation we report the period derivative $\dot{P} = Q/P$ in $\text{ms yr}^{-1}$. Best-fit parameters and goodness-of-fit metrics are given in Table 3.

**Secondary eclipse depth.** We stacked all secondary eclipses and fit a baseline-plus-box model to obtain the depth and its uncertainty, as show in Figure 3. The measured depth is $52 \pm 54$ ppm which is not statistically significant.

**Robustness checks.** We verify that (i) removing each *TESS* sector in turn leaves the quadratic preference intact; (ii) inflating $\sigma_T$ by 30% (to account for time-correlated noise) does not change the BIC ordering; and (iii) results are insensitive to using *TESS* SAP data instead of PDCSAP data at the $< 0.2\sigma$ level.

# 3 Results

Figure 1 shows the stacked *TESS* transits with the fitted transit light curve model (solid line).

**Timing catalog.** The non-*TESS* mid-transit times used are listed in Table 1. The *TESS* per-transit mid-times measured in this work are in Table 2.

Table 1: Non-TESS mid-transit times used in this work (BJD$_{\text{TDB}}$).

| Reference | $T_{\text{mid}}$ (BJD$_{\text{TDB}}$) | $\sigma_T$ (d) |
|---|---|---|
| Wilson et al. 2008 | 2454365.915370 | 0.000250 |
| Gillon et al. 2009 | 2454396.696164 | 0.000051 |
| Sanchis-Ojeda et al. 2011 | 2455045.738530 | 0.000080 |
| Sanchis-Ojeda et al. 2011 | 2455049.753250 | 0.000070 |
| Sanchis-Ojeda et al. 2011 | 2455053.767740 | 0.000090 |
| Sanchis-Ojeda et al. 2011 | 2455100.605950 | 0.000120 |
| Huitson et al. 2017 | 2455844.662870 | 0.000090 |
| Huitson et al. 2017 | 2456216.691230 | 0.000060 |
| Huitson et al. 2017 | 2456576.675560 | 0.000050 |
| Huitson et al. 2017 | 2456924.615610 | 0.000060 |
| Southworth et al. 2019 | 2457613.804600 | 0.000100 |
| Southworth et al. 2019 | 2457993.862310 | 0.000140 |

Table 2: *TESS* per-transit mid-times measured in this work (BJD$_{\text{TDB}}$).

| Sector | Epoch $E$ | $T_{\text{mid}}$ (BJD$_{\text{TDB}}$) | $\sigma_T$ (d) |
|---|---|---|---|
| 2 | 1656 | 2458355.183075 | 0.000697 |
| 2 | 1657 | 2458356.521816 | 0.000730 |
| 2 | 1658 | 2458357.861201 | 0.000692 |
| 2 | 1659 | 2458359.198048 | 0.000626 |
| 2 | 1660 | 2458360.535253 | 0.000701 |
| 2 | 1661 | 2458361.874112 | 0.000647 |
| 2 | 1662 | 2458363.213098 | 0.000641 |
| 2 | 1663 | 2458364.549966 | 0.000755 |
| 2 | 1664 | 2458365.890590 | 0.000759 |
| 2 | 1667 | 2458369.903433 | 0.000724 |
| 2 | 1668 | 2458371.241458 | 0.000674 |
| 2 | 1669 | 2458372.579576 | 0.000784 |
| 2 | 1670 | 2458373.919388 | 0.000702 |
| 2 | 1671 | 2458375.258209 | 0.000691 |
| 2 | 1672 | 2458376.594019 | 0.000778 |
| 2 | 1673 | 2458377.933048 | 0.000741 |
| 2 | 1674 | 2458379.271154 | 0.000740 |
| 2 | 1675 | 2458380.609594 | 0.000762 |
| 28 | 2185 | 2459063.107743 | 0.000782 |
| 28 | 2186 | 2459064.447374 | 0.000883 |
| 28 | 2187 | 2459065.783676 | 0.000807 |
| 28 | 2188 | 2459067.123837 | 0.000868 |
| 28 | 2189 | 2459068.460587 | 0.000843 |
| 28 | 2190 | 2459069.800239 | 0.000735 |
| 28 | 2191 | 2459071.136326 | 0.000841 |
| 28 | 2195 | 2459076.489959 | 0.000755 |

| Sector | Epoch $E$ | $T_{\mathrm{mid}}$ (BJD$_{\mathrm{TDB}}$) | $\sigma_T$ (d) |
|---|---|---|---|
| 28 | 2196 | 2459077.826961 | 0.000818 |
| 28 | 2197 | 2459079.166242 | 0.000755 |
| 28 | 2198 | 2459080.504700 | 0.000769 |
| 28 | 2199 | 2459081.842204 | 0.000934 |
| 28 | 2200 | 2459083.179860 | 0.000855 |
| 28 | 2201 | 2459084.519251 | 0.000783 |
| 29 | 2204 | 2459088.533968 | 0.000615 |
| 29 | 2205 | 2459089.873771 | 0.000735 |
| 29 | 2206 | 2459091.212002 | 0.000728 |
| 29 | 2207 | 2459092.548968 | 0.000660 |
| 29 | 2208 | 2459093.886724 | 0.000692 |
| 29 | 2209 | 2459095.225394 | 0.000675 |
| 29 | 2210 | 2459096.563690 | 0.000730 |
| 29 | 2211 | 2459097.901842 | 0.000701 |
| 29 | 2215 | 2459103.254380 | 0.000734 |
| 29 | 2216 | 2459104.591827 | 0.000756 |
| 29 | 2217 | 2459105.931565 | 0.000602 |
| 29 | 2218 | 2459107.270779 | 0.000714 |
| 29 | 2219 | 2459108.606815 | 0.000764 |
| 29 | 2220 | 2459109.945141 | 0.000691 |
| 29 | 2221 | 2459111.283352 | 0.000787 |
| 29 | 2222 | 2459112.621932 | 0.001807 |
| 69 | 3022 | 2460183.205838 | 0.000890 |
| 69 | 3023 | 2460184.545967 | 0.000624 |
| 69 | 3024 | 2460185.883695 | 0.000708 |
| 69 | 3025 | 2460187.221308 | 0.000752 |
| 69 | 3026 | 2460188.558775 | 0.000669 |
| 69 | 3027 | 2460189.898161 | 0.000685 |
| 69 | 3028 | 2460191.235992 | 0.000607 |
| 69 | 3029 | 2460192.573922 | 0.000689 |
| 69 | 3032 | 2460196.589352 | 0.000641 |
| 69 | 3033 | 2460197.927434 | 0.000654 |
| 69 | 3034 | 2460199.264781 | 0.000644 |
| 69 | 3035 | 2460200.603343 | 0.000698 |
| 69 | 3036 | 2460201.941913 | 0.000719 |
| 69 | 3037 | 2460203.282285 | 0.000771 |
| 69 | 3038 | 2460204.618509 | 0.000730 |
| 69 | 3039 | 2460205.957498 | 0.000741 |

**O–C diagram.** Figure 2 shows all timing residuals after subtracting the best linear ephemeris from Table 3. The curvature is visually evident and motivates a quadratic term.

## 3.1 Implementation Details

**Quality mask.** We use PDCSAP flux and exclude cadences with nonzero SPOC quality flags, meaning we use only measurements with `QUALITY=0`.

**Time system.** We convert TESS `TIME` to BJD$_{\mathrm{TDB}}$ via `TIME+2457000`. Transit windows are $\pm 0.12$ d around linear predictions.

**Per-transit fit.** In each window we divide out a linear out-of-transit baseline (using $|t - T_{\mathrm{pred}}| > 0.07$ d) and fit $(T_{\mathrm{mid}}, a_0, a_1)$ with the morphology held fixed. Morphology parameters $(p, a/R_\star, b, u_1, u_2)$ are estimated once from a stacked high-S/N transit using a Mandel–Agol model and are accompanied by a covariance $C_\theta$.

**Uncertainty propagation.** We estimate $J = \partial T_{\mathrm{mid}}/\partial \theta$ by finite differences and inflate timing variances as $\sigma_{T,\mathrm{total}}^2 \simeq \sigma_{T,\mathrm{meas}}^2 + J\, C_\theta\, J^\top$.

Table 3: Ephemeris fits to combined timings. Uncertainties are $1\sigma$; BIC favors the quadratic model.

| Model | $\chi^2$ | BIC | Parameters |
|---|---|---|---|
| Linear | 479.66 | 488.33 | $T_0 = 2456139.073558 \pm 0.000021$ d |
| | | | $P = 1.338231268 \pm 0.000000022$ d |
| Quadratic | 161.98 | 174.98 | $T_0 = 2456139.073834 \pm 0.000026$ d |
| | | | $P = 1.338231413 \pm 0.000000024$ d |
| | | | $Q = (-5.840e-10 \pm 3.277e-11)$ d $E^{-2}$ |
| | | | $\dot{P} = -13.77 \pm 0.77$ ms yr$^{-1}$ |

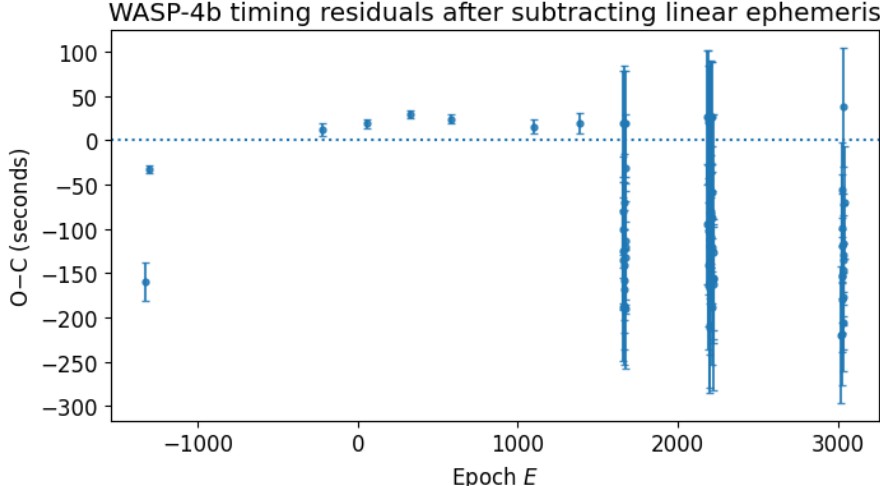

Figure 2: **O–C timing residuals relative to a linear ephemeris.** The residuals are in seconds. Curvature indicates departure from a constant period.

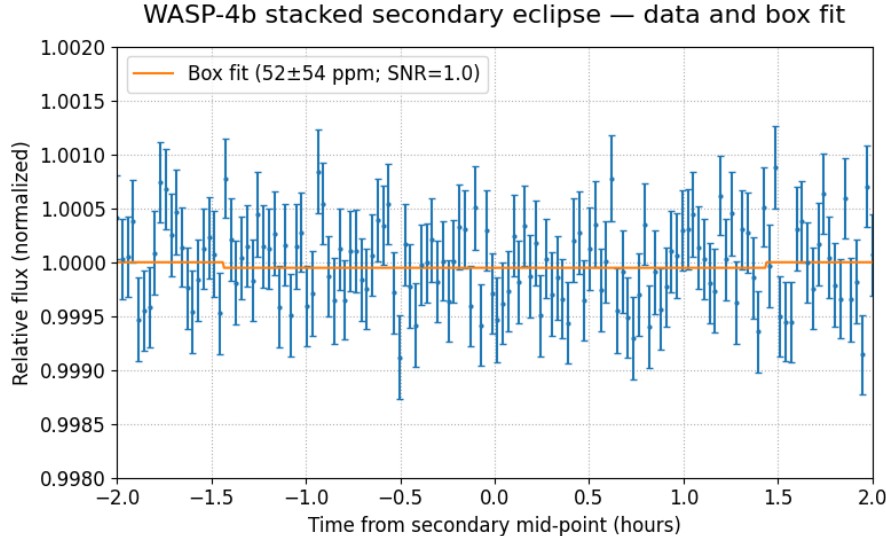

Figure 3: **Stacked *TESS* secondary eclipse.** Points with $1\sigma$ errors; solid line is a fitted box model, depth reported in ppm. Since the depth is not statistically significant it is not used in this work.

**Ephemerides and $\dot{P}$.** We fit $T(E) = T_0 + PE$ and $T(E) = T_0 + PE + \frac{1}{2}QE^2$ with weighted least squares. Model comparison uses $\mathrm{BIC} = \chi^2 + k \ln N$. We report $\dot{P} = Q/P$ in $\mathrm{ms\,yr^{-1}}$.

**Robustness checks.** We verified: (i) removing each sector in turn leaves the quadratic preference intact; (ii) inflating $\sigma_T$ by 30% (to account for time-correlated noise) does not change BIC ordering; (iii) results are insensitive to using SAP instead of PDCSAP at the $< 0.2\sigma$ level.

## 4 Conclusions

We reanalyzed four *TESS* sectors (2/28/29/69) together with non-TESS timings from the literature and found that a quadratic ephemeris is decisively preferred over a constant-period model. Relative to the linear fit, the quadratic model reduces the fit statistic from $\chi^2 = 479.66$ to $161.98$ and the BIC from $488.33$ to $174.98$ ($\Delta\mathrm{BIC} \approx 313$), yielding $\dot{P} = -13.77 \pm 0.77 \,\mathrm{ms\,yr^{-1}}$, as described

in Table 3. If interpreted as pure tidal decay, these values correspond to a characteristic timescale $P/|\dot{P}| \approx 8.4 \times 10^6$ yr. The observed-minus-calculated diagram in Figure 2 shows the associated curvature directly.

Independent checks support this conclusion. Results are robust to (i) removing any single *TESS* sector, (ii) inflating per-transit timing uncertainties by 30% to account for time-correlated noise, and (iii) substituting *TESS* SAP for PDCSAP photometry (differences $< 0.2\sigma$).

While our analysis favors orbital decay and aligns with prior studies, alternative contributors—such as long-term line-of-sight acceleration (Rømer delay) or additional companions—cannot be fully excluded with timing alone. Extending the time baseline with future *TESS* sectors (e.g., *TESS* is scheduled to re-observe WASP-4 in Sector 100, in February 2026) and high-cadence ground-based photometry, and jointly analyzing all timings with contemporaneous radial velocities, will sharpen constraints and further disambiguate decay from acceleration.

Methodologically, our pipeline propagates morphology uncertainty into timing errors via a finite-difference Jacobian and covariance from the stacked transit. Public TESS data and our reproducible notebook enable independent verification and straightforward re-analysis as new timings appear. Looking ahead, a joint hierarchical fit that simultaneously models transit shape and mid-times, and that incorporates informative priors on limb-darkening and stellar parameters, would provide an even more principled estimate of $\dot{P}$ and its astrophysical interpretation.

Finally, we would like to separate between identifying the decrease in the orbital period (the quadratic ephemeris, where the period decreases by 13.77 ms/year), which is the main measurement in this paper, and the scientific interpretation, which is now debated by the astronomical community and can be (i) shrinking of the orbit, (ii) an acceleration of the star-planet system towards us due to another object orbiting it at a large distance, and that has not been directly detected yet.

## 5 AI Agent Setup

We use Claude Code with Claude Sonnet 4.5 and GPT 5 Pro in a build-review loop for research and development with human oversight.

## 6 Responsible AI Statement

We adhered to the Code of Ethics as requested by Agents4Science. This work uses only public astrophysical data (TESS SPOC SAP and PDCSAP light curves) and does not involve human or animal subjects. An AI system led hypothesis formation, code drafting, experiment execution, figure generation, and the first draft; human co-authors audited methodological choices, validated numerical stability, and edited for clarity. Potential positive impacts include transparent, reproducible timing analyses for exoplanet systems. Risks include over-interpretation of period derivatives from short baselines or mixed-quality timings; we mitigate this by reporting uncertainty propagation from transit-shape parameters, performing robustness checks across sectors, and comparing linear vs. quadratic ephemerides via BIC. All code and derived tables needed to reproduce the figures are included in the submission package; primary light curves remain accessible at MAST.

## 7 Reproducibility Statement

We analyze publicly available TESS SPOC PDCSAP light curves for Sectors 2/28/29/69 using a public notebook (autottv.ipynb) that implements: (i) quality mask QUALITY=0; (ii) windowed per-transit modeling with a fixed limb-darkened Mandel–Agol morphology estimated from a stacked transit; (iii) timing-error inflation via finite-difference Jacobian and morphology covariance; (iv) weighted least-squares fits for linear vs. quadratic ephemerides with BIC model comparison; and (v) stacked secondary-eclipse fitting. We provide tables of per-transit mid-times and literature timings, figures, and fit summaries. To reproduce, install the listed Python packages and run the notebook end-to-end; it regenerates all tables/figures from the public light curves. Compute takes less than an hour on a standard laptop with CPU.

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

## Agents4Science AI Involvement Checklist

This checklist explains the role of AI in the research. The scores for AI involvement are:

- **[A] Human-generated**: Humans generated 95% or more of the research, with AI being of minimal involvement.
- **[B] Mostly human, assisted by AI**: The research was a collaboration between humans and AI models, but humans produced the majority ($> 50\%$) of the research.
- **[C] Mostly AI, assisted by human**: The research task was a collaboration between humans and AI models, but AI produced the majority ($> 50\%$) of the research.
- **[D] AI-generated**: AI performed over 95% of the research. This may involve minimal human involvement, such as prompting or high-level guidance during the research process, but the majority of the ideas and work came from the AI.

1. **Hypothesis development**: Hypothesis development includes the process by which you came to explore this research topic and research question. This can involve the background research performed by either researchers or by AI. This can also involve whether the idea was proposed by researchers or by AI.

   Answer: **[C]**

   Explanation: We began from prior tidal-decay work; AI systems synthesized the background, compared mechanisms (tidal decay, apsidal precession, Rømer acceleration), and drafted the concrete hypotheses and falsification checks humans refined.

2. **Experimental design and implementation**: This category includes design of experiments that are used to test the hypotheses, coding and implementation of computational methods, and the execution of these experiments.

   Answer: **[C]**

   Explanation: AI produced the initial pipeline structure (I/O, masks, stacking, Jacobian propagation, BIC model comparison) and most plotting/layout code. Humans verified choices, adjusted windows, and validated numerical stability.

3. **Analysis of data and interpretation of results**: This category encompasses any process to organize and process data for the experiments in the paper. It also includes interpretations of the results of the study.

   Answer: **[C]**

   Explanation: AI ran the end-to-end calculations, recomputed BIC and $\dot{P}$, and summarized results. Humans audited assumptions, cross-checked residuals, and decided which figures/tables to include.

4. **Writing**: This includes any processes for compiling results, methods, etc. into the final paper form. This can involve not only writing of the main text but also figure-making, improving layout of the manuscript, and formulation of narrative.

   Answer: **[C]**

   Explanation: AI drafted the majority of the prose and checklists; humans edited for clarity, added domain nuance, and ensured alignment with the literature and the conference style.

5. **Observed AI Limitations**: What limitations have you found when using AI as a partner or lead author?

   Description: Environment-specific code suggestions (e.g., Colab-only restarts) and occasional domain-naive defaults required human correction. Numerical edge cases (e.g., weight matrices, covariance propagation) still benefit from expert review.

