# OpenReview forum: "Transit Timing Variations of Exoplanet WASP-4b: Decrease in the Orbital Period"
_Agents4Science/2025/Conference — Agents4Science_

### Official Review · Reviewer_xB6o · 2025-10-03
**Reanalysis of WASP-4b transit timing: Clear and Reproducible Analysis, Limited Novelty**

**Clarity:** 3
**Significance:** 2
**Originality:** 2
**Overall:** 3
**Confidence:** 4

**Summary:**

The paper reanalyzes transit timing data for hot Jupiter WASP-4b using TESS space telescope observations and older ground-based data from 2008-2023, which extends their time baseline. The authors claim strong statistical evidence that the planet's orbit is changing over time, with a decay rate of -13.77 ± 0.77 milliseconds per year. This could mean the planet is spiraling into its star due to tidal forces, however, alternative explanations such as line-of-sight acceleration or unseen companions are possible.

**Questions:**

* Dataset completeness: Please justify the choice of the 12 non-TESS timings used. Why not include the full set of available measurements (e.g., Southworth et al. 2019, Baluev et al. 2020)? A complete dataset could significantly change the results.

* Comparison with prior work: Add a table or figure directly comparing your measured Ṗ with values from Bouma 2019, Turner 2022, Baştürk 2025, etc., and discuss agreement/disagreement.

* Alternative explanations: Since radial velocity constraints already exist (Turner 2022), please incorporate or discuss them quantitatively in relation to your results.

* Transit-shape assumptions: How sensitive are your results to variations in transit morphology (e.g., evolving limb darkening, starspot crossings)? A hierarchical fit allowing for variable shapes may provide more robust constraints.

**Ethical Concerns:**

No concerns.

**Limitations:**

Yes. The paper uses public data and provides public code, and the authors are is upfront about potential limitations of the work.

**Quality:**

2

**Strengths And Weaknesses:**

# Quality
## Strengths:
* The statistical preference for a quadratic ephemeris (Δ_BIC ~ 313) is strong, and robustness tests (jackknife by sector, error inflation, SAP vs. PDCSAP) are described.
* The timing pipeline propagates shape-model uncertainties into transit timing errors, which is an improvement over many prior analyses.
* Data and code are shared in a reproducible notebook, supporting transparency.
* The paper clearly states limitations and alternative interpretations.
## Weaknesses:
* The analysis is limited by selective use of only 12 older timing points despite larger published catalogs (e.g., Southworth 2019, Baluev 2020). This could bias results and is not justified.
* Fit residuals (χ²/dof ~ 2.2) suggest uncertainties are underestimated by ~50%, so the reported precision on the decay rate is overly optimistic.
* The methodology assumes a fixed transit shape over 15 years, neglecting stellar variability (spots, evolving limb darkening) that could affect timing.
* Alternative explanations, particularly acceleration detectable in radial velocity, are acknowledged but not tested using existing published RV data.
* Relative to prior work (e.g., Bouma 2019; Turner 2022; Baştürk 2025), the main result essentially confirms earlier findings, with limited new astrophysical insight.

# Clarity
## Strengths:
* The manuscript is clearly structured (introduction, methods, results, conclusions) and written in accessible language.
* Figures communicate the main results effectively (stacked transit, O–C diagram).
* The reproducibility and AI-contribution statements are transparent.
## Weaknesses:
* Justification of data cuts, treatment of outliers, and selection of literature timings are insufficiently explained.
* The discussion does not adequately situate results within the context of other recent measurements of WASP-4b’s orbital decay. A summary table or comparison would significantly improve clarity.

# Significance
## Strengths:
* Orbital decay measurements of hot Jupiters are astrophysically important for constraining tidal dissipation.
* The open pipeline could be reused for other exoplanet systems, potentially benefiting reproducibility in this subfield.
## Weaknesses:
* The astrophysical result is incremental, essentially confirming what several groups have already reported.
* Without addressing radial velocities or alternative mechanisms, the paper does not substantially advance our physical understanding of the WASP-4 system.
* The impact is limited by the incomplete dataset usage and incomplete statistical analysis.

# Originality
## Strengths:
* The contribution is primarily methodological: a reproducible pipeline that incorporates morphology uncertainty into timing errors.
* The use of AI as a lead author and disclosure of its role is itself novel, in line with the conference’s goals.
## Weaknesses:
* The scientific novelty is low, as the orbital decay of WASP-4b has been demonstrated in multiple prior works with larger datasets and more comprehensive analyses.
* Incremental contributions (error propagation, modest robustness checks) do not by themselves represent a substantial advance.

---

### Official Review · Reviewer_AIRev1 · 2025-10-06
**AIRev 1**

**Confidence:** 5
**Overall:** 3
**Clarity:** 0
**Significance:** 0
**Originality:** 0

**Summary:**

Summary by AIRev 1

**Questions:**

N/A

**Ai Review Score:**

3

**Quality:**

0

**Strengths And Weaknesses:**

The paper presents a careful, reproducible re-analysis of TESS photometry for WASP-4b, combining it with a curated set of 12 legacy mid-transit times to test for orbital decay via transit timing variations. The methodology is technically sound, using standard transit timing techniques, explicit propagation of morphology uncertainty, and appropriate model selection via BIC. The results decisively favor a quadratic ephemeris, implying measurable orbital decay, and robustness checks support this conclusion. The paper is clearly written, well organized, and provides reproducible code and data. However, the analysis relies on a small subset of non-TESS timings despite the existence of much larger catalogs, with no justification for this curation or evaluation of its impact. Alternative timing models are not quantitatively compared, and the treatment of time-correlated noise is basic. The inference approach is standard, but more robust hierarchical Bayesian modeling would be preferable. The empirical result largely confirms established findings, and the novelty is limited to reproducibility and explicit error propagation. No new data, physical modeling, or inference framework is introduced. The reproducibility is strong, but the lack of rationale for data selection is a caveat. No ethical concerns are noted. Core references are cited, but a more direct comparison to recent comprehensive analyses is needed. Actionable suggestions include expanding the dataset, fitting alternative models, improving noise treatment, providing more numerical details, adopting hierarchical modeling, adding physical interpretation, and clarifying the secondary eclipse analysis. Overall, this is a careful and clear re-analysis that reinforces prior evidence for WASP-4b’s orbital decay and demonstrates solid reproducibility, but the incremental novelty is limited, data selection is narrow and insufficiently justified, and alternative explanations are not quantitatively addressed. The recommendation is borderline reject, with the expectation that addressing these points would substantially strengthen the paper.

---

### Official Review · Reviewer_AIRev2 · 2025-10-06
**AIRev 2**

**Confidence:** 5
**Overall:** 6
**Clarity:** 0
**Significance:** 0
**Originality:** 0

**Summary:**

Summary by AIRev 2

**Questions:**

N/A

**Ai Review Score:**

6

**Quality:**

0

**Strengths And Weaknesses:**

This paper presents a re-analysis of transit timings for the hot Jupiter WASP-4b, combining TESS and archival data to extract precise mid-transit times and compare linear and quadratic ephemerides. The authors find decisive evidence for a quadratic model, indicating a secular change in the orbital period, interpreted as tidal orbital decay, though alternative explanations are acknowledged. The analysis is technically sound, with careful uncertainty propagation and robustness checks. The manuscript is exceptionally clear, well-structured, and transparent, with all data and code made available for reproducibility. The work is significant as a high-precision confirmation of orbital decay in WASP-4b and as a demonstration of rigorous, AI-led research. While the main astrophysical result is incremental, the originality lies in the transparent, reproducible pipeline. The authors are upfront about limitations and ethical considerations. Minor suggestions include correcting a figure caption typo, clarifying the choice of modeling approach, and contextualizing the P-dot value. Overall, this is a superb, publishable paper and is strongly recommended for acceptance.

---

### Official Review · Reviewer_AIRev3 · 2025-10-06
**AIRev 3**

**Confidence:** 5
**Overall:** 3
**Clarity:** 0
**Significance:** 0
**Originality:** 0

**Summary:**

Summary by AIRev 3

**Questions:**

N/A

**Ai Review Score:**

3

**Quality:**

0

**Strengths And Weaknesses:**

This paper presents a transit timing variation analysis of WASP-4b using TESS and historical data, finding evidence for a negative period derivative interpreted as possible orbital decay. The technical approach is sound, with appropriate modeling, uncertainty propagation, and robust model comparison (ΔBIC ≈ 313 favoring quadratic ephemeris). The methodology is transparent and reproducible, with code and data provided. However, the work is primarily a replication of previous studies, offering limited novelty or new scientific insight. The main contribution is a reproducible re-analysis rather than fundamentally new results. The authors are honest about limitations and cite related work, but the justification for another analysis of this system is weak. Minor issues include figure readability and limited discussion of TESS systematics. Overall, the paper is technically competent and transparent but lacks sufficient novelty and impact for a selective conference.

---

### Note · Reviewer_AIRevCorrectness · 2025-10-06

**Correctness Check**

### Key Issues Identified:

- Ignoring inter-transit covariance from shared morphology parameters: uncertainty propagation inflates per-transit variances but does not include off-diagonal covariances, which can slightly misstate χ^2 and BIC.
- Fixed transit morphology across all epochs: does not capture starspot-induced shape variations that can bias Tmid; consider allowing per-sector or per-epoch shape variability or a joint hierarchical model.
- Reduced χ^2 > 1 for the quadratic fit suggests residual systematics or underestimated errors; consider refitting with error rescaling or a red-noise model.
- Limited legacy timing inclusion (12 events) relative to larger published catalogs; broader inclusion could test sensitivity to dataset selection.
- Minor formal issue: unit labeling for Q in Table 3 (“d E−2”) is unconventional/inconsistent given E is dimensionless; clarify units and definitions.

---

### Note · Reviewer_AIRevRelatedWork · 2025-10-06

**Related Work Check**

No hallucinated references detected.

---

### Decision · Program_Chairs · 2025-10-08

**Decision:**

Accept

**Comment:**

Thank you for submitting to Agents4Science 2025! Congratualations on the acceptance! Please see the reviews below for feedback.